# Description and genome-wide analysis of *Profundicola chukchiensis* gen. nov., sp. nov., marine bacteria isolated from bottom sediments of the Chukchi Sea

**Lyudmila Romanenko[1], Nadezhda Otstavnykh[1], Valeriya Kurilenko[1], Peter Velansky[2], Viacheslav Eremeev[1], Valery Mikhailov[1], Marina P. Isaeva[1]***

1 G.B. Elyakov Pacific Institute of Bioorganic Chemistry, Far Eastern Branch, Russian Academy of Sciences, Vladivostok, Russia, 2 A.V. Zhirmunsky National Scientific Center of Marine Biology, Far Eastern Branch, Russian Academy of Sciences, Vladivostok, Russia

* issaeva@gmail.com

## Abstract

Two Gram-negative, aerobic halophilic non-motile strains designated KMM 9713 and KMM 9724$^T$ were isolated from the bottom sediments sampled from the Chukchi Sea in the Arctic Ocean, Russia. The novel strains grew in 0.5−5% NaCl, at 7−42˚C, and pH 5.5−10.5. Phylogenetic analyses based on 16S rRNA gene and whole genome sequences revealed that strains KMM 9713 and KMM 9724$^T$ were close to each other and shared the highest 16S rRNA gene sequence similarity of 91.28% with the type strain *Ornithobacterium rhinotracheale* DSM 15997$^T$ and 90.15–90.92% with the members of the genus *Empedobacter* in the family *Weeksellaceae*. Phylogenetic trees indicated that strains KMM 9713 and KMM 9724$^T$ formed a distinct line adjacent to their relative *O. rhinotracheale* DSM 15997$^T$. The average nucleotide identity values between strain KMM 9724$^T$ and *O. rhinotracheale* DSM 15997$^T$, *Empedobacter brevis* NBRC 14943$^T$, and *Moheibacter sediminis* CGMCC 1.12708$^T$ were 76.73%, 75.78%, and 74.65%, respectively. The novel strains contained the predominant menaquinone MK-6 and the major fatty acids of iso-C$_{17:0}$ 3-OH, iso-C$_{15:0}$ followed by iso-C$_{17:1}\omega$6. Polar lipids consisted of phosphatidylethanolamine, one an unidentified aminophospholipid, two unidentified aminolipids, and two or three unidentified lipids. The DNA G+C contents of 34.5% and 34.7% were calculated from genome sequence of the strains KMM 9713 and KMM 9724$^T$, respectively. Based on the phylogenetic evidence and distinctive phenotypic characteristics, strains KMM 9713 and KMM 9724$^T$ are proposed to be classified as a novel genus and species *Profundicola chukchiensis* gen. nov., sp. nov. The type strain of *Profundicola chukchiensis* gen. nov., sp. nov. is strain KMM 9724$^T$ (= KACC 22806$^T$).

## Introduction

The peculiarity of the Chukchi Sea among the Arctic seas is its high-latitude location, which is reflected in the presence of ice cover most of the time during the year and low average annual

**Data Availability Statement:** The type strain of Profundicola chukchiensis gen. nov. sp. nov. is

strain KMM 9724T (=KACC 22806T). The DDBJ/ENA/GenBank accession numbers for the 16S rRNA gene and the whole-genome shotgun sequences of strains KMM 9724T and KMM 9713 are OP604014 and LC379507, and JANAIE010000000.1 and JANCMU010000000.1, respectively.

**Funding:** This work was supported by a grant from the Ministry of Science and Higher Education, Russian Federation 15.BRK.21.0004 (Contract No. 075-15-2021-1052). The funders had no role in study design, data collection and analysis, decision to publish, or preparation of the manuscript.

**Competing interests:** The authors have declared that no competing interests exist.

water temperature. Even in summer, the temperature of water layers deeper than 10–12 m remains almost at zero values. The bottom of the Chukchi Sea is flat, the average depth of the continental shelf is 50–60 meters, and the depth of shoals is 20–30 m. Therefore, studies of microorganisms dwelling in the arctic marine sediments provide insight into their genetic capabilities to live in extreme habitat conditions.

The members of the family *Flavobacteriaceae* [1−3] of the phylum *Bacteroidota* constitute one of the dominant bacterial groups which have been reported to be widespread microorganisms inhabiting marine environments [4, 5]. The family *Flavobacteriaceae* contains a large number of species and genera while the phylogenetic relationships on the basis of the 16S rRNA gene sequences between some of them were not completely resolved until recently [6]. Phylogenomic studies based on whole genome sequencing analysis of 1000 type strains of the phylum *Bacteroidota* (formerly *Bacteroidetes*) have shown that the family *Flavobacteriaceae* is non-monophyletic and therefore should be divided [7]. As a result, a new family *Weeksellaceae* has been proposed to include the genera *Algoriella*, *Apibacter*, *Bergeyella*, *Chishuiella*, *Chryseobacterium*, *Cloacibacterium*, *Cruoricaptor*, *Elizabethkingia*, *Empedobacter*, *Moheibacter*, *Ornithobacterium*, *Riemerella*, *Wautersiella*, and *Weeksella* as the type genus that formed a separate clade from the type genus *Flavobacterium* of the family *Flavobacteriaceae* [7]. At the time of writing, the family *Weeksellaceae* comprises 15 genera with correct and validly published names (https://lpsn.dsmz.de/family/weeksellaceae, accessed on 15 May 2023).

During a study on the bacterial biodiversity of the bottom sediments of the Chukchi Sea in the Arctic Ocean, two Gram-negative, aerobic, yellowish-pigmented, non-motile bacteria, KMM 9713 and KMM 9724T were recovered and investigated by using phenotypic and molecular methods; the results obtained are reported in this study. Phylogenetic analyses demonstrated that strains KMM 9713 and KMM 9724T formed a distinct lineage within the family *Weeksellaceae* adjacent to the type bacterium *Ornithobacterium rhinotracheale* DSM 15997T. Based on the phylogenomic analyses data and distinctive phenotypic characteristics, a novel genus and species *Profundicola chukchiensis* gen. nov., sp. nov. is described to accommodate the novel marine isolates KMM 9713 and KMM 9724T.

## Materials and methods

### Bacterial strains

Strains KMM 9713 and KMM 9724T were isolated from a deep bottom sediment sampled at a depth of 29 m from the Chukchi Sea (70˚59.60′ N, 177˚35.8′ W, near Wrangel Island), Russia, during the expedition of R/V *Academician Oparin*, in September 2016, as described previously [8]. The novel bacteria were cultivated aerobically on Marine Agar 2216 (MA) or in Marine Broth (MB) 2216 (BD Difco) at 28˚C and stored at −70˚C in MB 2216 supplemented with 20% (*v/v*) glycerol. The type strain KMM 9724T has been deposited in the Collection of Marine Microorganisms (KMM), G.B. Elyakov Pacific Institute of Bioorganic Chemistry, Far Eastern Branch, Russian Academy of Sciences, Vladivostok, Russia, and in the Korean Agricultural Culture Collection (KACC), Agricultural Microbiology Division, National Academy of Agricultural Science, Korea, as KACC 22806T. The type strain *Empedobacter tilapiae* KCTC 62904T was purchased from the Korean Collection for Type Cultures, KCTC, Korea, to be used in the comparative phenotypic tests.

### Phenotypic characterization

Gram-staining, oxidase, catalase reactions, and motility (the hanging drop method) were determined as described by Gerhardt et al. [9]. Gliding motility was observed according to the method of Bowman [10]. The morphology of cells grown on MA 2216 and negatively stained

with a 1% (*w/v*) phosphotungstic acid on carbon-coated 200-mesh copper grids was examined by electronic transmission microscopy Libra 120 FE (Carl Zeiss), provided by the Far Eastern Centre of electronic microscopy, A.V. Zhirmunsky National Scientific Center of Marine Biology, Far Eastern Branch, Russian Academy of Sciences. Hydrolysis of starch, casein, gelatin, Tween 80, DNA, L-tyrosine, chitin, and growth at different salinities (0–12% (*w/v*) NaCl), temperatures (5–45˚C), and pH values (4.5–10.5) were carried out using artificial seawater (ASW) as early described [11, 12]. Biochemical tests were performed using API 20E, API 20NE, API ID32 GN, and API ZYM test kits (bioMérieux, Marcy-l'Étoile, France) as described by the manufacturer except the cultures were suspended in ASW. Antibiotic susceptibility was examined using commercial paper discs (Research Centre of Pharmacotherapy, St. Petersburg, Russia) impregnated with the following antibiotics (mg per disc, unless otherwise indicated): ampicillin (10), benzylpenicillin (10 U), vancomycin (30), gentamicin (10), kanamycin (30), carbenicillin (100), chloramphenicol (30), neomycin (30), oxacillin (10), oleandomycin (15), lincomycin (15), ofloxacin (5), rifampicin (5), polymyxin (300 U), streptomycin (30), cephazolin (30), cephalexin (30), erythromycin (15), nalidixic acid (30), and tetracycline (30).

## Chemotaxonomic analyses

Strains KMM 9713 and KMM 9724[T] and the type strain *Empedobacter tilapiae* KCTC 62904[T] were grown on MA 2216 at 30˚C. Lipids were extracted using the method of Folch et al. [13]. Two-dimensional thin layer chromatography of polar lipids was carried out on Silica gel 60 F$_{254}$ (10 x 10 cm, Merck, Germany) using chloroform-methanol-water (65:25:4, *v/v*) for the first direction, and chloroform-methanol-acetic acid-water (80:12:15:4, *v/v*) for the second one [14] and spraying with specific reagents [15]. Fatty acid methyl esters (FAMEs) were prepared according to the procedure of the Microbial Identification System (MIDI) [16]. The analysis of FAMEs was performed using the GC-2010 chromatograph (Shimadzu, Kyoto, Japan) equipped with capillary columns (30 m x 0.25 mm I.D.), one coated with Supecowax-10 and the other with SPB-5. Identification of FAMEs was accomplished by equivalent chain length values and comparing the retention times of the samples to those of standards. In addition, FAMEs were analyzed using a GC-MS Shimadzu QP2020 (column Shimadzu SH-Rtx-5MS, the temperature program from 160˚C to 250˚C, at a rate of 2˚C/min). Menaquinones fraction was isolated using liquid column chromatography on silica gel. The lipid extract in chloroform was applied to the column, and the neutral lipid fraction with menaquinones was washed off with three column volumes of chloroform. Analysis was performed using GC-MS with SH-Rtx-5ms column, the temperature was programed from 200˚C to 240˚C, (10˚C/min), then from 240˚C to 325˚C, (3˚C/min) and kept for 30 min at 325˚C. The injector temperature was 300˚C, mass spectrometer scan range 50–1000 m/z. The presence of flexirubin pigments was investigated as described by Fautz and Reichenbach [17].

## 16S rRNA gene sequencing and phylogenetic analysis

Genomic DNAs of strains KMM 9713 and KMM 9724[T] were extracted using a commercial genomic DNA extraction kit (Fermentas, EU) following the manufacturer's instruction. The 16S rRNA genes were PCR-amplified and sequenced as described in a previous paper [18]. The 16S rRNA gene sequences of the strains KMM 9713 and KMM 9724[T] (1393 and 1401 bp, respectively) were compared with those of the closest relatives using the BLAST (http://www.ncbi.nlm.nih.gov/blast/, accessed on 15 May 2023) and EzBioCloud service [19]. Model testing and phylogenetic analysis were conducted using Molecular Evolutionary Genetics Analysis (MEGA X, version 10.2.1) [20]. Phylogenetic trees were constructed by the neighbor-joining and the maximum-likelihood methods, and the distances were calculated according to the

Kimura two-parameter model [21]. The robustness of phylogenetic trees was estimated by the bootstrap analysis of 1000 replicates.

## Whole-genome sequencing, phylogenomic, and comparative analyses

The genomic DNAs were obtained from the strains KMM 9713 and KMM 9724[T] using the High Pure PCR Template Preparation Kit (Roche, Basel, Switzerland). The quantity and quality of the genomic DNA was measured using DNA gel electrophoresis and the Qubit 4.0 Fluorometer (Thermo Fisher Scientific, Singapore, Singapore). The DNA sequencing libraries were prepared using Nextera DNA Flex kits (Illumina, San Diego, CA, USA) and subsequently sequenced using paired-end (2 x 150 bp) runs on an Illumina MiSeq platform. The reads were trimmed using Trimmomatic version 0.39 [22] and their quality assessed using FastQC version 0.11.8 (https://www.bioinformatics.babraham.ac.uk/projects/fastqc/, accessed on 21 August 2021). Contigs assembled with SPAdes version 3.15.3 [23] were used to calculated genome metrics with QUAST version 5.0.2 [24]. The genome completeness and contamination were estimated by CheckM version 1.1.3 [25] based on the taxonomic-specific workflow (lineage *Flavobacteriales*).

The draft genome assemblies were annotated using NCBI Prokaryotic Genome Annotation Pipeline (PGAP) and Rapid Annotation using Subsystem Technology (RAST) [26, 27]. Comparisons of the Average Nucleotide Identity (ANI), Average Amino Acid Identity (AAI), and digital DNA-DNA hybridization (dDDH) values of the strains KMM 9713 and KMM 9724[T] and their closest neighbors were performed with the online server ANI/AAI-Matrix [28] and TYGS platform [29], respectively. The phylogenomic analysis was performed using PhyloPhlAn software version 3.0.1 based on a set of 400 conserved bacterial protein sequences [30].

Genome-wide analysis of orthologous clusters and pairwise genome comparisons between genomes were performed using OrthoVenn2 (https://orthovenn2.bioinfotoolkits.net/home, accessed on 22 December 2022) [31]. Annotation of secondary metabolite biosynthetic gene clusters was conducted using antiSMASH server version 6.1.1 (https://antismash.secondarymetabolites.org/#!/start, accessed on 25 September 2022). Annotation of bacterial protein secretion systems and related appendages in the genomes was conducted with MacSyFinder (TXSScan) program on Galaxy server Version 2.0+galaxy2 (https://galaxy.pasteur.fr/, accessed on 01 December 2022). A conserved C-terminal domain (CTD) of the type IX secretion system (T9SS) was identified using website Pfam 35.0 (http://pfam-legacy.xfam.org/search#tabview=tab1). To identify carbohydrate-active enzymes (CAZymes) the dbCAN2 meta server version 10 was used with default settings (http://cys.bios.niu.edu/dbCAN2, accessed on 14 June 2022) [32]. Predictions by two of the three algorithms integrated within the server were considered sufficient for CAZy family assignments. Additionally, the genomes were annotated using dbCAN3 meta server (http://cys.bios.niu.edu/dbCAN2, accessed on 28 April 2023) and corresponding output files are presented in S1 File. The unique genes of strains KMM 9713 and KMM 9724[T] were functional annotated using eggNOG-mapper v2 server (http://eggnog-mapper.embl.de/, accessed on 12 January 2023) [33]. The relative abundances of CAZymes and distribution of unique genes according to COG categories were visualized by heat maps using the pheatmap version 1.0.12 package in RStudio version 2022.02.0 +443 with R version 4.1.3.

## Nucleotide sequence accession number

The 16S rRNA gene sequence and genome sequence of strains KMM 9724[T] and KMM 9713 were deposited in GenBank/EMBL/DDBJ under the accession numbers OP604014 and LC379507, and JANAIE010000000.1 and JANCMU010000000.1, respectively. Strain KMM

9724[T] was deposited in the Korean Agricultural Culture Collection (KACC) under the number of KACC 22806[T].

# Results and discussion

## Phylogenetic and phylogenomic analyses

The average nucleotide identity (96.9%) and DNA-DNA hybridization (74.4%) values obtained on the basis of whole genome sequence comparison between two novel strains KMM 9713 and KMM 9724[T] confirmed their assignment to the same species. Comparative 16S rRNA gene sequence analysis showed that the novel strains KMM 9713 and KMM 9724[T] belong to the family *Weeksellaceae* (phylum *Bacteroidota*) and their close phylogenetic neighbors were found to be *Ornithobacterium rhinotracheale* DSM 15997[T] with 90.22−91.28% gene sequence similarity and members of the genus *Empedobacter* sharing 90.15−90.92% sequence similarity. The low values of 16S rRNA gene sequence similarities with the members of related genera demonstrated that strains KMM 9713 and KMM 9724[T] can be representatives a novel genus in the family *Weeksellaceae*. The phylogenetic trees generated by the different algorithms (Fig 1) placed the novel strains KMM 9713 and KMM 9724[T] as a separate lineage adjacent to *O. rhinotracheale* DSM 15997[T]. In addition, the neighbor-net network (S1 Fig) based on concatenated nucleotide sequences from MLSA data confirmed the clustering of both novel strains and their divergence from *O. rhinotracheale* DSM 15997[T]. Finally, the phylogenetic tree based on concatenated 400 translated protein sequences from whole genome sequences supported the position of the novel strains as a distinct line adjacent to *O. rhinotracheale* DSM 15997[T] in the family *Weeksellaceae* (Fig 2).

The ANI and dDDH values between strain KMM 9724[T] and *O. rhinotracheale* DSM 15997[T], *Empedobacter brevis* NBRC 14943[T], and *Moheibacter sediminis* CGMCC 1.12708[T] were 76.73%, 75.78%, and 74.65%, and values of 20.5%, 18.9%, and 21.5%, respectively (Fig 3A, 3C). These values obtained were significantly lower than the ANI and dDDH values of 95−96% and 70%, respectively, which have been accepted for bacterial species discrimination [34]. The AAI values between genomes of both novel strains and related bacteria *O. rhinotracheale* DSM 15997[T], *E. brevis* NBRC 14943[T], and *M. sediminis* CGMCC 1.12708[T] ranged from 60.28% to 52.73% (Fig 3B). These values fall into the range of AAI values between 45% and 65% proposed by Konstantinidis et al. [35] to delineate bacterial genera. The phylogenomic analysis data evidence that the strains KMM 9713 and KMM 9724[T] do not belong to any of recognized genera and could be classified as an individual genus and species of the family *Weeksellaceae*.

## Genomic characteristics and comparative analysis

The whole genome sequences of strains KMM 9724[T] and KMM 9713 were determined using Illumina MiSeq platform. Both genomes were obtained at high completeness (99.01%) without contamination according to the CheckM evaluation. The 16S rRNA sequences extracted from the genomes were identical to those obtained by PCR amplification. The draft genomes were de novo assembled into 108 and 47 contigs, with a N50 values of 440,336 and 448,720 bp, a L50 values of 3 and 2, respectively. The genome sizes were estimated at 2.62 and 2.67 Mbp in length with coverage of 180 X and 140 X, respectively. The genome sequences were in accordance with the proposed minimal standards for the bacterial taxonomy [34]. The genome sequences contain a total of 2,366 and 2,348 genes, 37 and 38 tRNAs, and 3 rRNA genes (one each of 5S, 16S, and 23S).

Based on RAST annotation, the genome of type strain KMM 9724[T] showed the presence of 220 functional subsystems, of which the largest number of annotated genes, about 49%, were

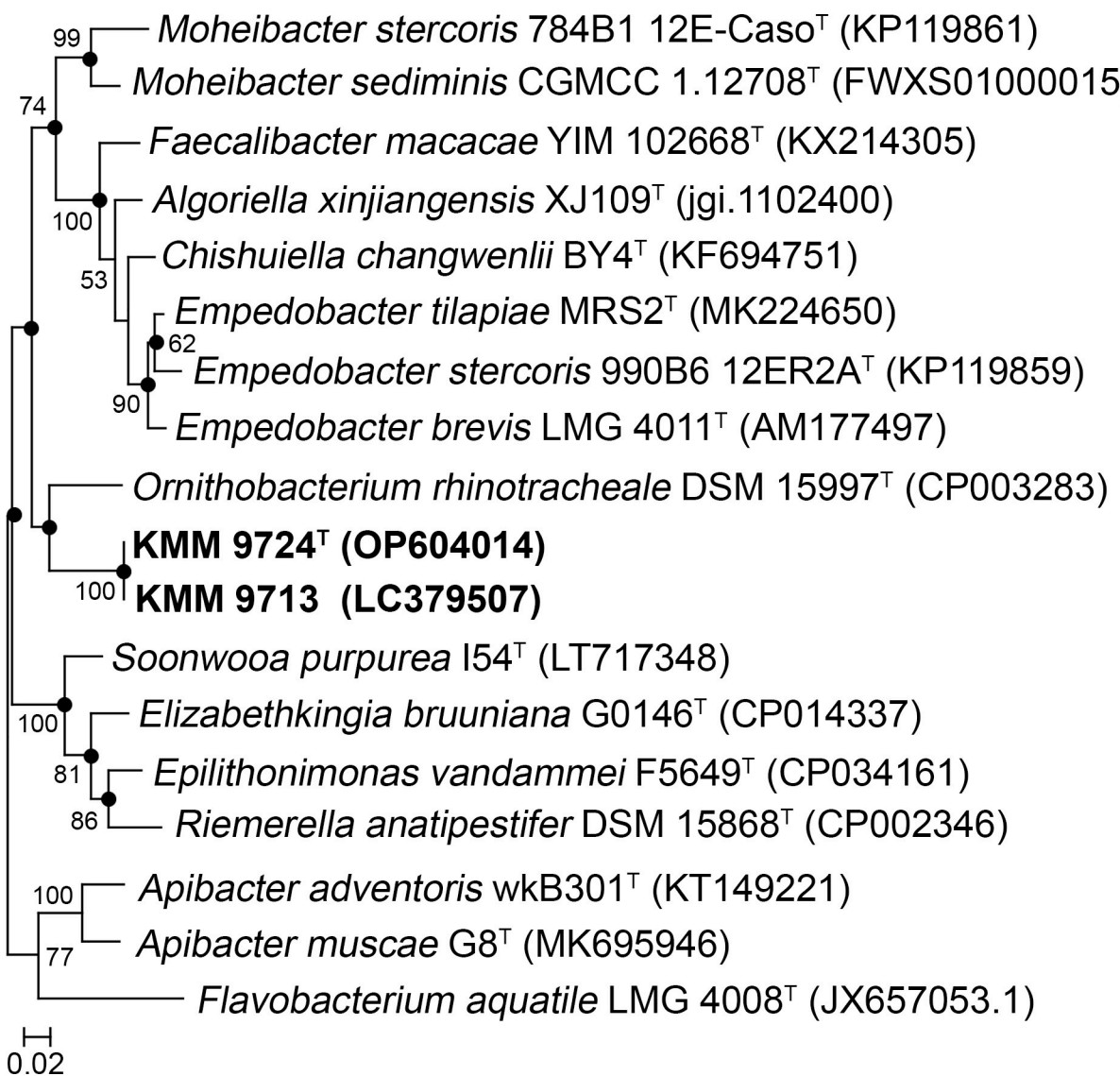

**Fig 1. Maximum-likelihood tree based on 16S rRNA gene sequences available from the GenBank/EMBL/DDBJ databases (accession numbers are given in parentheses) showing relationships of the novel strains KMM 9724$^T$, KMM 9713, and related members of the family *Weeksellaceae*.** Filled circles indicate the corresponding nodes that were observed in the neighbor-joining tree. Numbers indicate bootstrap values as percentage greater than 50. These values are based on 1000 replicates. Bar, 0.02 substitutions per nucleotide position.

assigned to three subsystems: "Protein Metabolism" (149 genes), "Amino Acids and Derivatives" (135 genes), and "Cofactors, Vitamins, Prosthetic Groups, Pigments" (119 genes). Among peptidases for protein degradation were predicted aminopeptidases (EC 3.4.11.-), metallocarboxypeptidases (EC 3.4.17.-), dipeptidase (EC 3.4.13.-), serine endopeptidase (EC 3.4.21.-), omega peptidase (EC 3.4.19.-), and ATP-dependent proteases. According to the annotation, the strain possesses an antibiotic- (fluoroquinolones) and metal-resistance genes (Cu, Co), and genes for biosynthesis of biotin. The antiSMASH analysis showed the genome encodes one secondary metabolite cluster for terpene biosynthesis. The genome of another strain KMM 9713 showed a similar distribution of genes to functional subsystems with slight differences.

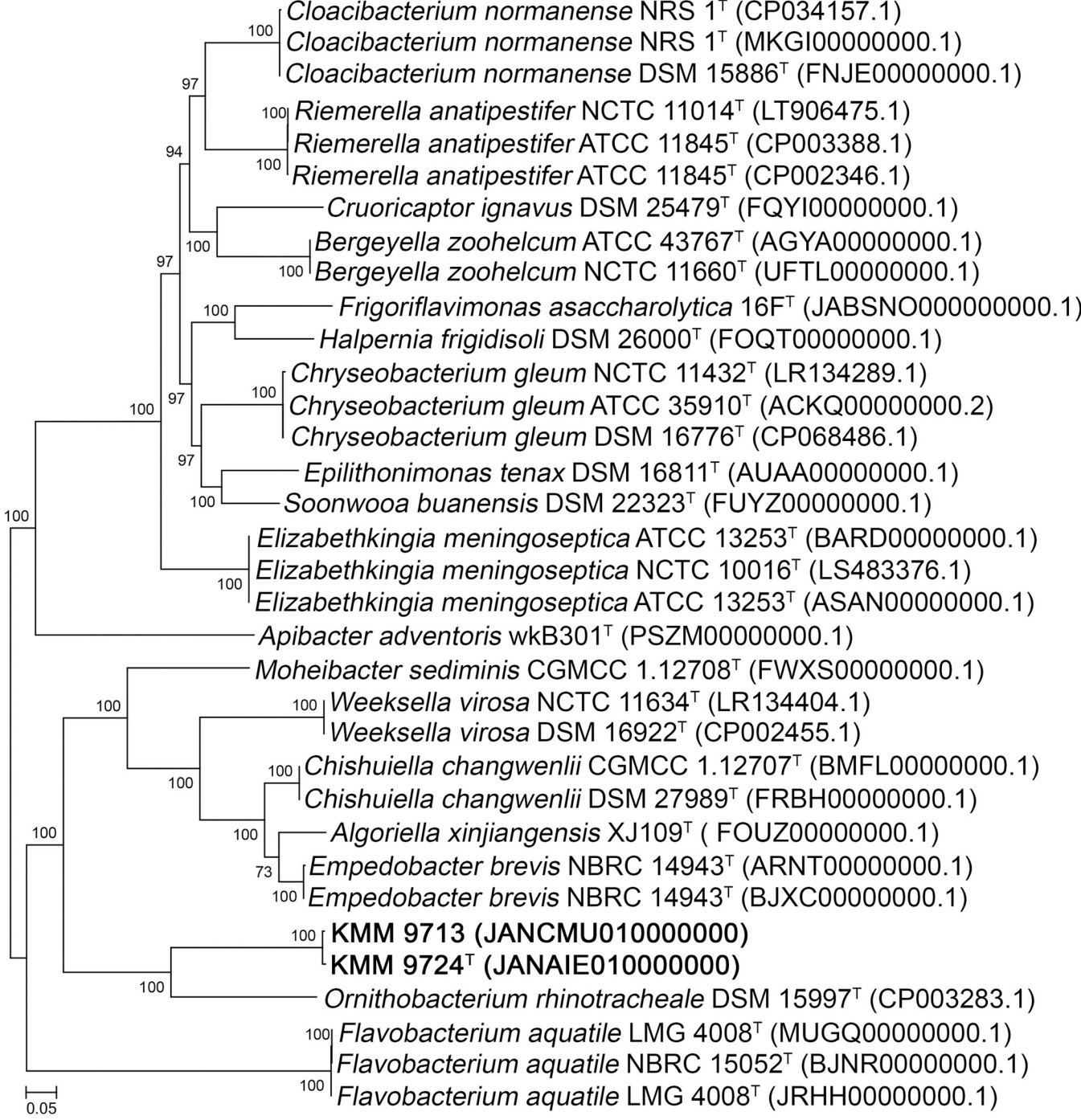

**Fig 2. Maximum-likelihood tree based on concatenated 400 protein sequences showing phylogenetic position of the novel strains KMM 9724ᵀ and KMM 9713 and related members of the family *Weeksellaceae*.** Bootstrap values are based on 100 replicates. *Flavobacterium aquatile* was used as an outgroup. Bar, 0.05 substitutions per amino acid position.

Annotation of bacterial protein secretion systems in the genomes was conducted with Mac-SyFinder (TXSScan) program. It was shown that strains KMM 9724ᵀ and KMM 9713 have all mandatory and accessory genes for the type IX secretion system (T9SS), such as *gldK*, *gldL*, *gldM*, *gldN*, *porV*, *sprA*, *sprE*, *sprT*, and *gldJ*, *porU*, *porQ*, respectively. Interestingly, the T9SS is

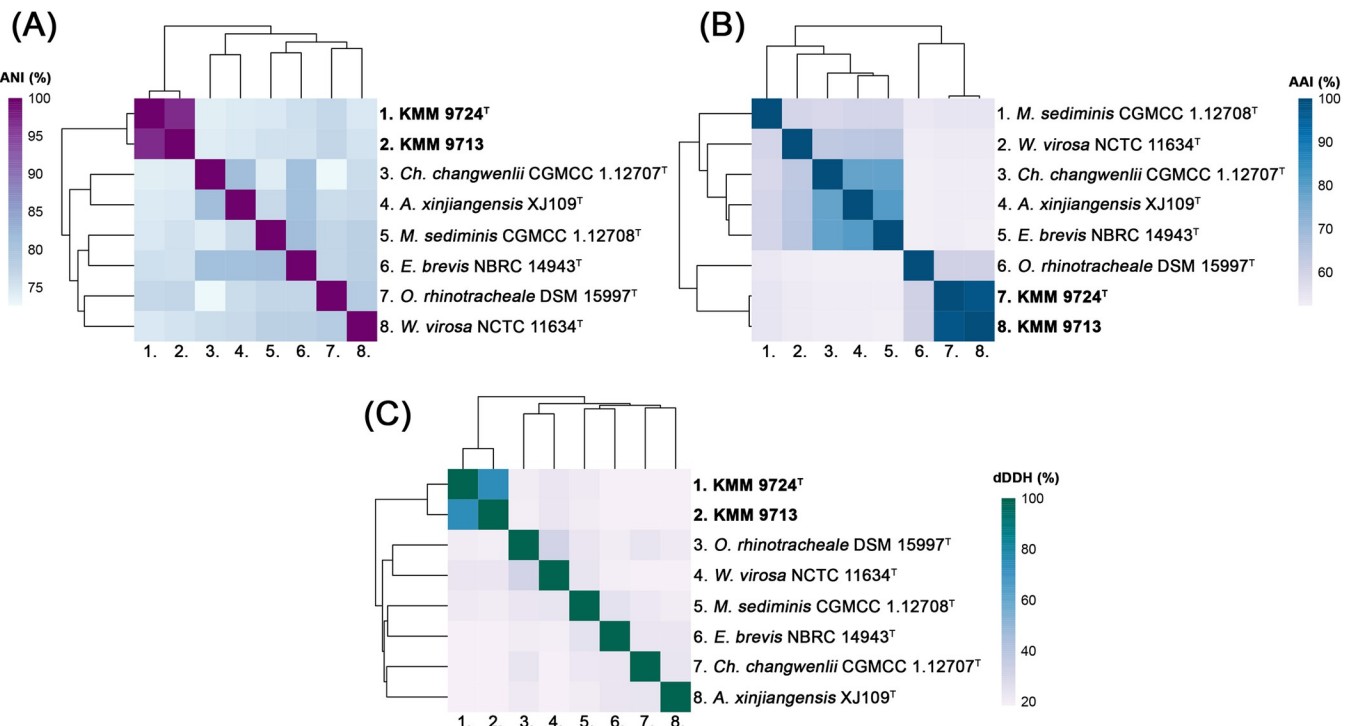

**Fig 3.** The heatmaps of the genome relatedness indexes display pairwise values of: (A) average nucleotide identities (ANI); (B) average amino acid identities (AAI); (C) digital DNA-DNA hybridization (dDDH) in percentages calculated using online servers ANI/AAI-Matrix [28] and TYGS [29], respectively.

exclusively present in the majority of families of the phylum *Bacteroidota* and also required for gliding motility of bacterial cells, though it was first detected in the non-motile human pathogen *Porphyromonas gingivalis* [36, 37]. As known, T9SS substrates are cell-associated proteins that contain a C-terminal domain (CTD) and are involved in virulence, gliding motility, and the degradation of complex biopolymers [38]. Among predicted substrates of the T9SS strains KMM 9724[T] and KMM 9713 possess a set of endonucleases, peptidases S8 and M1, reprolysins, adhesins, Cu-binding proteins, and other Por_Secre_tail (pfam: PF18962.3) containing proteins. This may imply that the strains are able to consume complex biopolymers of sediments in the Chukchi Sea as a specific metabolic strategy. Another secretion system identified in the genomes was type I secretion system (T1SS), which is widespread in Gram-negative bacteria. It provides a one-step secretion of substrates across two membranes without any periplasmic intermediate into the extracellular space [39]. The genomes of the KMM 9724[T] and KMM 9713 encode from two to five copies of each mandatory gene *abc*, *mfp*, and *omf*.

Whole-genome comparative analysis of novel strains and representatives of six related genera from the same clade on the phylogenomic tree (Fig 2) was performed using orthologous clustering with OrthoVenn2 server. According to the pairwise genome comparisons (Fig 4A), the strains KMM 9724[T] and KMM 9713 share the most of orthologous protein clusters of 3167 with each other, while those values between studied strains with type strains of genera *Algoriella*, *Chishuiella*, *Moheibacter*, *Weeksella*, *Empedobacter*, and *Ornithobacterium* were from 2705 to 2711. However, despite on an almost equal distribution of shared clusters among strains KMM 9724[T] and KMM 9713, and representatives of related genera, their phylogenetically closest relative genus is *Ornithobacterium*.

To clarify genus-related features, the orthologous clustering analysis of strains KMM 9724[T], KMM 9713, and *O. rhinotracheale* DSM 15997[T] was conducted (Fig 4B). The analysis revealed

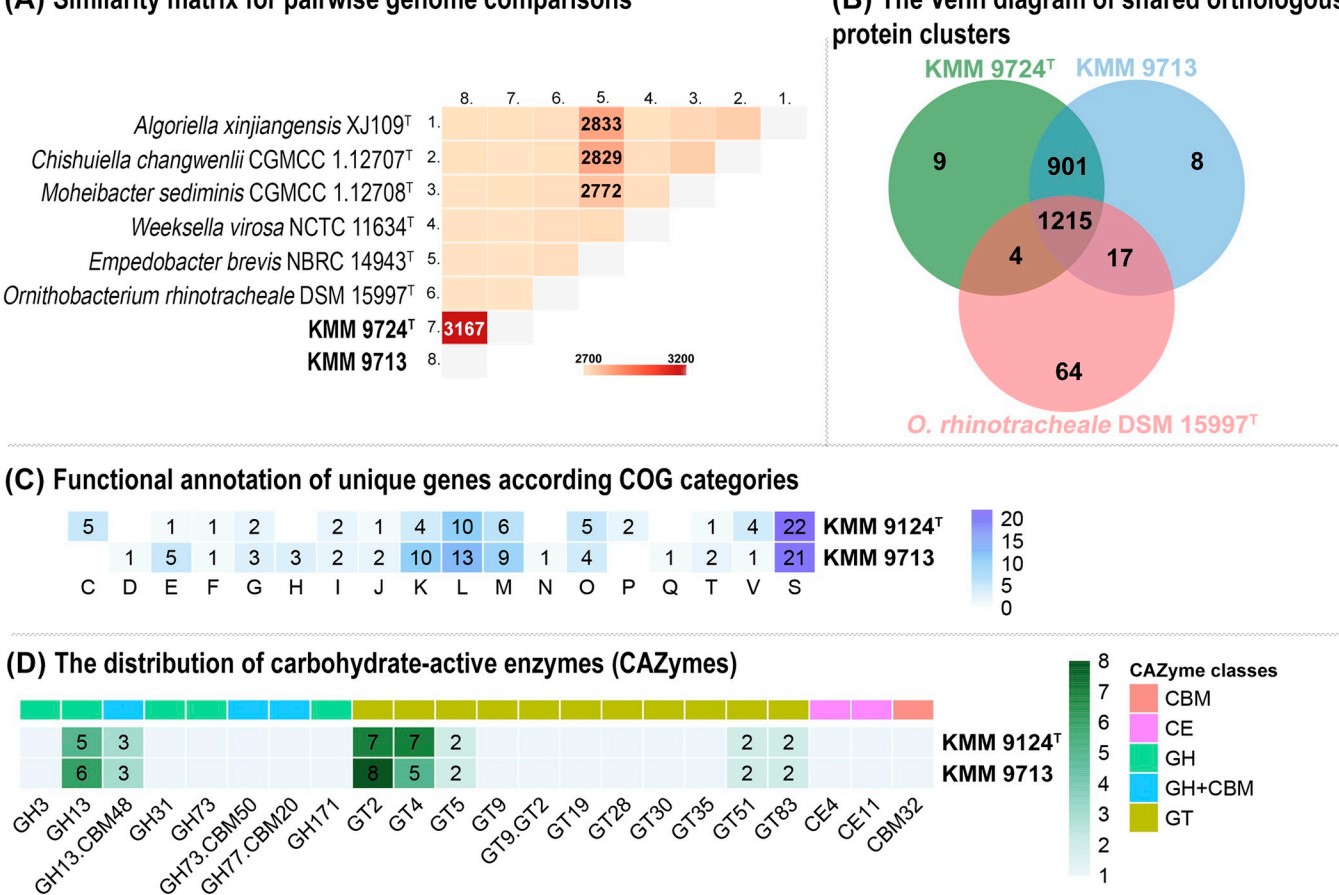

**Fig 4. Whole-genome comparative analysis of novel strains.** (A) The pairwise heatmap shows overlapping orthologous protein clusters between any pair of genomes. (B) The Venn diagram plotted by OrthoVenn2 shows shared orthologous protein clusters between the genomes of strains KMM 9124[T] and KMM 9713 with their closest phylogenetic relative. The numbers of shared and unique gene clusters are shown. (C) Functional annotation of unique genes of the strains KMM 9124[T] and KMM 9713. (D) The distribution of carbohydrate-active enzymes (CAZymes) in genomes of KMM 9124[T] and KMM 9713. The heat map shows the number of genes assigned to individual CAZyme families. GH, glycoside hydrolases; GT, glycosyltransferases, CE, carbohydrate esterases; CBM, carbohydrate-binding modules.

that strains form 2218 orthologous gene clusters (gene families), of which 1011 orthologous clusters (at least contains two strains) and 1207 single-copy gene clusters. It was shown that presumably genus-specific genes of the strain *O. rhinotracheale* DSM 15997[T] are represented by 64 clusters of orthologs formed by 178 paralogous genes and 665 single-copy genes. The strains KMM 9724[T] and KMM 9713 share 901 clusters of 1808 genes, including paralogs, which are likely genus- and species-specific. A close inspection of these genes revealed that most of genes were assigned to following biological processes: biological processes (GO:0008150, 11.3%), metabolic processes (GO:0008152, 10.9%), cellular metabolic processes (GO:0044237, 8.8%), and nitrogen compound metabolic processes (GO:0006807, 6%). Among gene ontology annotations of molecular function activities of transferases, transporters, oxido-reductases, and hydrolases were enriched. In addition to shared species-specific genes, inter-strain differences between KMM 9724[T] and KMM 9713 were also identified. As such, the type strain KMM 9724[T] has 9 unique clusters of 22 paralogous genes and 220 single-copy genes, and strain KMM 9713 has 8 unique clusters of 19 paralogous genes and 211 single-copy genes.

To elucidate intra-species differences the unique single-copy genes of the strains were functional annotated via eggNOG-mapper v2 server (Fig 4C). Genes were assigned to the following COG categories: C, energy production and conversion; D, cell cycle control and mitosis; E, amino acid metabolism and transport; F, nucleotide metabolism and transport; G, carbohydrate metabolism and transport; H, coenzyme metabolism, I, lipid metabolism; J, translation; K, transcription; L, replication and repair; M, cell wall/membrane/envelop biogenesis; N, cell motility; O, post-translational modification, protein turnover, chaperone functions; P, inorganic ion transport and metabolism; Q, secondary structure; T, signal transduction; V, defense mechanisms; S, function unknown. According to the COG classes annotation of these unique genes, the most abundant functional classes in both strains were "transcription", "replication and repair", and "cell wall/membrane/envelop biogenesis".

Apparently, these genes may be responsible for adaptation to extreme conditions and let this species to inhabit the Chukchi Sea in the Arctic Ocean. It can be concluded that metabolism is mainly carried out by consumption protein-containing substrates rather than carbohydrates, since both strains KMM 9724$^T$ and KMM 9713 encode a few periplasmic glycoside hydrolases (Fig 4D). Nevertheless, strains of novel species of the novel genus may be a source of biotechnological relevant pullulanases (GH13), peptidases and proteases.

## Phenotypic characterization and chemotaxonomy

Bacteria KMM 9713 and KMM 9724$^T$ were observed to be Gram-negative, aerobic, catalase- and oxidase-positive, non-motile. They formed yellowish colonies with regular edges of 1−3 mm in diameter on MA 2216. Electron microscopy images of the cells depicted ovoid or rod-shaped morphology (Fig 5). The cell dimensions were 0.5−0.75 μm in width and 1.2−3.5 μm in length. Capsular material around cells can be produced.

The novel strains required sodium ions for growth and grew in the narrow salinity range of 0.5 −5% (*w/v*) NaCl and at a temperature between 7˚C and 42˚C. The novel bacteria were not able to hydrolyse a broad number of polymeric substrates (Table 1) and assimilate most carbon sources in API 32GN, API 20E, and API 20NE tests. Cultural, physiological, and biochemical characteristics of the novel bacteria are given in Table 1 and in the genus and species descriptions.

Strains KMM 9724$^T$ and KMM 9713 contained the major menaquinone MK-6 (89.1–78.8%) with small amounts of MK-5 (7.5–17.5%) and a trace of MK-4 (3.4–3.7%). Major fatty acids were found to be iso-$C_{17:0}$ 3-OH, iso-$C_{15:0}$ followed by iso-$C_{17:1}\omega6$ (Table 2). The predominance of MK-6 and iso-$C_{17:0}$ 3-OH, iso-$C_{15:0}$ is characteristic for the members of the family *Weeksellaceae* [7]. Fatty acid profiles of strains KMM 9724$^T$ and KMM 9713 and related *O. rhinotracheale* and *Empedobacter* bacteria were similar in a large proportion of iso-$C_{15:0}$ and iso-$C_{17:0}$ 3-OH, while *O. rhinotracheale* strains contained significant amounts of iso-$C_{15:0}$ and iso-$C_{17:0}$ 3-OH (>70%) (Table 2). Bacteria KMM 9713 and KMM 9724$^T$ differed in content of anteiso-$C_{15:0}$, anteiso-$C_{15:0}$ 3-OH, and anteiso-$C_{17:0}$ 3-OH, which were not detected in related *O. rhinotracheale* and *Empedobacter* bacteria. In addition, summed $C_{16:1}\omega7c$/ $C_{16:1}\omega6c$ occurred only in the fatty acid profiles of *Empedobacter* (Table 2).

Polar lipids of novel strains consisted of phosphatidylethanolamine (PE), one an unidentified aminophospholipid (APL), two unidentified aminolipids (AL), and two unidentified lipids (L) (S2 Fig). Strain KMM 9713 contained additionally one unidentified lipid. Strains KMM 9724$^T$ and KMM 9713 were similar in their polar lipid composition with dominant component of PE to those reported for *O. rhinotracheale*, *Empedobacter*, and other members of the family *Weeksellaceae* [7]. The DNA G+C contents of 34.5–34.7% were calculated from the whole genome sequences of the strains KMM 9713 and KMM 9724$^T$. These values fall into the range of 29.2–45.6 mol% found for the members of the family *Weeksellaceae* [7] and are close to

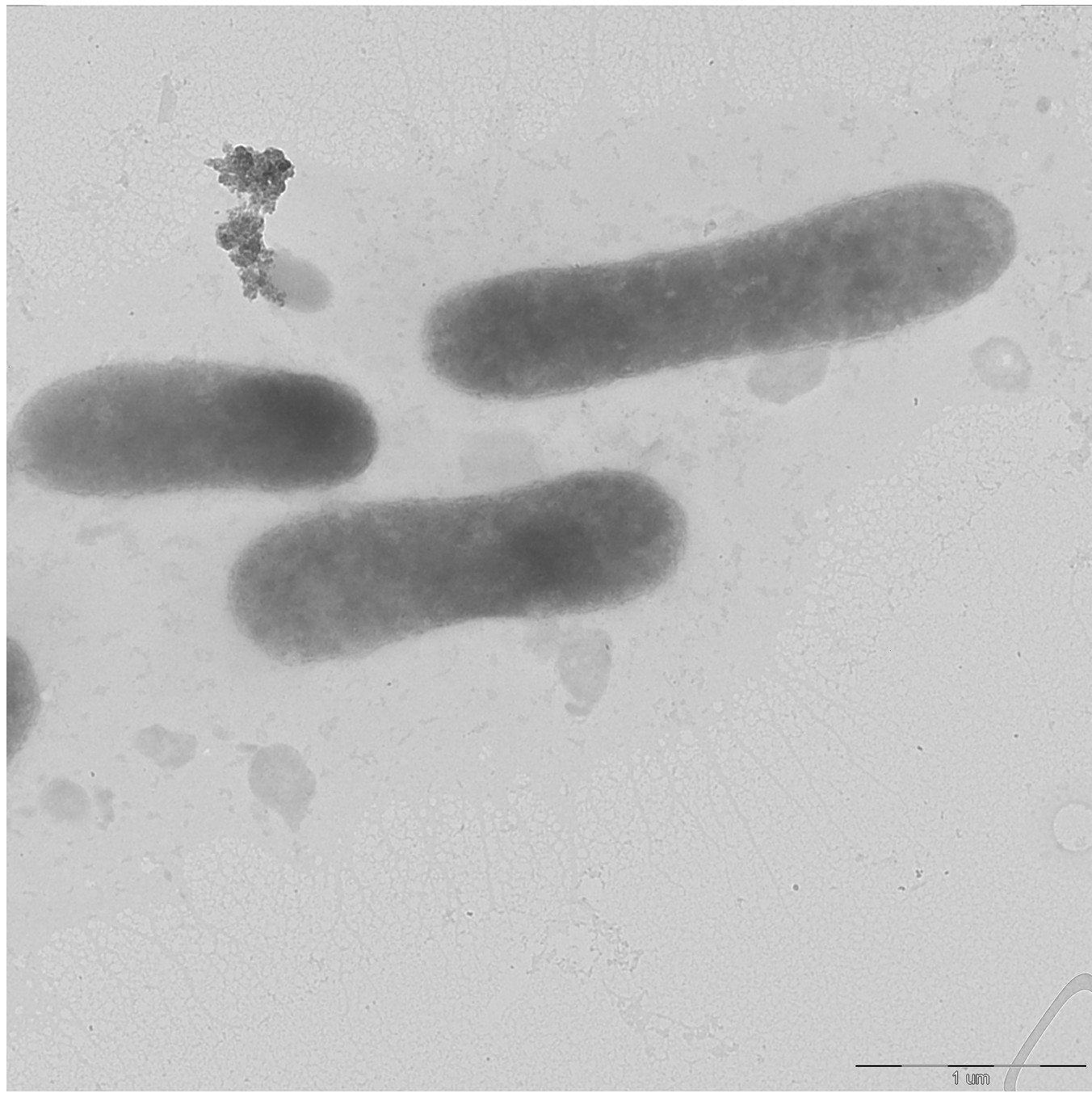

**Fig 5. A transmission electron micrograph of strain KMM 9724<sup>T</sup> cells grown in MB 2216 for 48 h.** Bar, 1 μm.

those of *Empedobacter* members (31.7–33.6 mol%) but lower than DNA G+C means (37–39 mol%) of *O. rhinotracheale* (Table 1).

The phylogenetic relationships observed on the basis of 16S rRNA gene and whole genome sequences, and genetic distinctness as revealed by ANI and dDDH analyses were supported by phenotypic differences of the novel isolates KMM 9713 and KMM 9724$^T$ in their growth temperature and salinity ranges, enzymes activity and substrate hydrolysis. Differential phenotypic

**Table 1. Differential characteristics of strains KMM 9724[T], KMM 9713, and related genera *Ornithobacterium* and *Empedobacter*.**

| Feature | 1 | 2 | 3 |
|---|---|---|---|
| Colonies pigmentation | Yellowish | Non-pigmented | Cream, yellow, or pale yellow |
| Flexirubin-type pigment | – | – | V(+) |
| Catalase | + | – | + |
| Growth at: 37˚C | + | + | V(+) |
| 42˚C | + | + | – |
| Arginine dihydrolase | – | + | + |
| Hydrolysis of: | | | |
| Gelatin | + | – | V(–) |
| Esculin | + | – | V(+) |
| Starch | – | + | V(+) |
| Enzyme activity: | | | |
| Leucine arylamidase | – | + | + |
| Valine arylamidase | – | + | + |
| Cystine arylamidase | – | + | + |
| Trypsin | – | + | V(+) |
| α-Chymotrypsin | – | ND | V(+) |
| α-galactosidase | – | + | – |
| β-galactosidase | – | + | – |
| α-glucosidase | – | + | V(+) |
| N-acetyl-β-glucosaminidase | – | + | – |
| β-glucuronidase | – | – | V(–) |
| Major fatty acids | iso-$C_{17:0}$ 3-OH, iso-$C_{15:0}$, iso-$C_{17:1}\omega6$ | iso-$C_{15:0}$, iso-$C_{15:0}$ 3-OH, iso-$C_{17:0}$ 3-OH | $C_{16:1}\omega7c$/$C_{16:1}\omega6c$, iso-$C_{17:0}$ 3-OH iso-$C_{15:0}$, $C_{16:0}$ |
| DNA G+C content (mol%) | 34.7–34.5 | 37–39 | 31.7–33.6 |
| Isolation source | Marine bottom sediment | Respiratory tracts of birds | Water samples, clinical material, cow manure, fish intestine |

Taxa: 1, KMM 9724[T], KMM 9713 [data from present study]; 2, *O. rhinotracheale* [40]; 3, *Empedobacter* [41–43]. All strains are non-motile, positive for alkaline phosphatase, esterase (C4), esterase lipase (C8), acid phosphatase, and negative for nitrate reduction, lipase C 14, β-glucosidase, α-mannosidase, and α-fucosidase. Symbols: +, positive; –, negative; V (–), variable reaction between species, reaction of the type strain of the type species is negative; V (+), variable reaction between species, reaction of the type strain of the type species is positive.

characteristics are indicated in Table 1. Based on the combined phylogenetic evidence, phenotypic and biochemical characteristics, it is proposed to classify strains KMM 9713 and KMM 9724[T] as a novel genus and species, *Profundicola chukchiensis* gen. nov., sp. nov. with the type strain of the type species KMM 9724[T] (= KACC 22806[T]).

## Conclusions

### Description of *Profundicola chukchiensis* gen. nov.

*Profundicola* (Pro.fun.di.co'la. L. neut. adj. *profundum*, the deep-sea; L. masc./fem. n. suff. -*cola*, inhabitant; N.L. masc. n. *Profundicola*, inhabitant of the deep-sea.

Cells are Gram-negative, non-motile, non-spore-forming and rod-shaped. Aerobic. The major fatty acids are iso-$C_{17:0}$ 3-OH, iso-$C_{15:0}$. The major polar lipid is phosphatidylethanolamine. The predominant menaquinone is MK-6. Isolated from the marine environment. Phylogenetically the genus *Profundicola* belongs to the family *Weeksellaceae* in the phylum *Bacteroidota*. The type species is *Profundicola chukchiensis* with the type strain KMM 9724[T].

**Table 2. Fatty acid (%) composition of strains KMM 9724$^T$, KMM 9713, and related bacteria of the genera *Ornithobacterium* and *Empedobacter*.**

| Fatty acid | 1 | 2 | 3 | 4 | 5 | 6 | 7 |
|---|---|---|---|---|---|---|---|
| $C_{14:0}$ | Tr | Tr | Tr | Tr | 1.3 | 1.2 | 1.3 |
| iso-$C_{15:0}$ | 11.9 | 13.5 | 57.4+6.1 | 15.4 | 14.7 | 15.3 | 13.2 |
| anteiso-$C_{15:0}$ | 7.9 | 9.6 | Tr | Tr | – | – | – |
| $C_{15:1}$ $\omega6$ | Tr | Tr | – | 1.7 | 2.2 | 1.1 | Tr |
| $C_{15:0}$ | Tr | Tr | – | 1.6 | – | – | – |
| iso-$C_{16:1}\omega6$ | Tr | Tr | – | 1.8 | 1.5* | 4.5* | 1.4* |
| iso-$C_{16:0}$ | 6.2 | 4.9 | – | 3.7 | 3.3 | 10.9 | 5.0 |
| $C_{16:1}\omega7c$/ $C_{16:1}\omega6c$ | Tr | Tr | – | 6.5 | 20.2 | 14.0 | 17.6 |
| $C_{16:1}\omega5c$ | 4.6 | 3.7 | – | 5.1 | 1.4 | 2.7 | 8.3 |
| anteiso-$C_{15:0}$ 2-OH | 1.7 | 2.1 | – | 1.0 | – | – | – |
| $C_{16:0}$ | Tr | 1.5 | 2.9+1.3 | 4.0 | 14.0 | 8.1 | 5.3 |
| iso-$C_{15:0}$ 3-OH | 4.9 | 3.9 | 8.1+1.9 | 5.4 | 3.8 | 3.0 | 3.8 |
| anteiso-$C_{15:0}$ 3-OH | 3.7 | 4.2 | – | – | – | – | – |
| iso-$C_{17:1}\omega9$ | – | – | – | – | 2.3 | 1.6 | 1.5 |
| iso-$C_{17:1}\omega7$ | Tr | Tr | – | 1.5 | | | |
| iso-$C_{17:1}\omega6$ | 9.2 | 10.9 | – | 5.3 | | | |
| iso-$C_{17:1}$/anteiso-$C_{7:1}$ B | | | | | 2.9 | 3.7 | 6.3 |
| iso-$C_{17:0}$ | 1.0 | 1.8 | 1.5+1.0 | Tr | 2.6 | 1.5 | Tr |
| $C_{16:0}$ $\omega7,8$-Me (Cyclo) | 3.0 | 3.4 | – | – | – | – | – |
| $C_{16:0}$ 3-OH | – | – | 2.8+1.8 | - | 6.4 | 4.0 | 5.6 |
| iso-$C_{16:0}$ 3-OH | 9.3 | 6.0 | – | 5.7 | 3.9 | 11.7 | 5.2 |
| anteiso-$C_{16:0}$ 3-OH | 1.3 | 1.0 | – | 6.0 | – | – | – |
| iso-$C_{17:0}$ 3-OH | 16.9 | 16.4 | 20.2+5.0 | 19.6 | 13.3 | 12.0 | 17.4 |
| anteiso-$C_{17:0}$ 3-OH | 8.5 | 7.9 | – | – | – | – | – |

* In iso-$C_{16:1}$ double bond position in unknown.

Taxa: 1, KMM 9724$^T$; 2, KMM 9713 [data from present study]; 3, *O. rhinotracheale* [40]; 4, *E. tilapiae* KCTC 62904$^T$ [data from present study]; 5; *E. brevis* KACC 11960$^T$; 6, *E. stercoris* LMG 28501$^T$; 7, *E. falsenii* CCUG51536$^T$ [42]. Fatty acids representing <1% in all strains tested were not shown; Tr, trace amounts (<1%). Symbols: –, not detected.

## Description of *Profundicola chukchiensis* sp.nov.

*Profundicola chukchiensis* (chuk.chi.en'sis. N.L. masc./fem. adj. *chukchiensis*, from the Chukchi Sea, the isolation place of the type strain).

In addition to properties given in the genus description the species is characterized as follows: cells are ovoid or rod-shaped 0.5−0.75 μm in width and 1.2−3.5 μm in length. Capsular material can be observed. Non-motile. On MA 2216 produces yellowish-pigmented smooth colonies with the regular edges of 1−3 mm in diameter. Oxidase- and catalase-positive. Require NaCl for growth. Growth occurs in the presence of 0.5−5% (*w/v*) NaCl (optimum, 2−3% NaCl) and at 7−42˚C (optimum, 28−30˚C). The pH range for growth is 5.5−10.5 (optimum, 6.5−7.5). Positive for hydrolysis of DNA and $H_2S$ production from thiosulfate. Negative for nitrate reduction and hydrolysis of casein, starch, chitin, and Tween 80. In the API 20E positive for indole production and gelatin hydrolysis; and negative for ONPG, arginine dihydrolase, lysine decarboxylase, ornithine decarboxylase, citrate utilization, $H_2S$ and urease production under anaerobic conditions, tryptophane deaminase, acetoin production (Voges-Proskauer reaction), and oxidation/fermentation of D-sucrose, D-glucose, D-mannitol, inositol, D-sorbitol, L-rhamnose, D-melibiose, amygdalin, and L-arabinose.

According to the API 20NE, positive for indole production, esculin and gelatin hydrolysis; and negative for nitrate reduction, PNPG test, glucose fermentation, arginine dihydrolase, urease, assimilation of D-glucose, D-mannitol, maltose, D-gluconate, L-malate assimilation of D-mannose, L-arabinose, N-acetylglucosamine, caprate, adipate, citrate, and phenylacetate.

In the API ZYM tests positive for alkaline phosphatase, esterase (C4), esterase lipase (C8), leucine arylamidase, acid phosphatase, and naphthol-AS-BI-phosphohydrolase, and negative for lipase (C14), valine arylamidase, cystine arylamidase, trypsin, α-chymotrypsin, α-galactosidase, β-galactosidase, β-glucosidase, α-glucosidase, β-glucuronidase, N-acetyl-β-glucosaminidase, α-mannosidase, and α-fucosidase.

Susceptible to (content per disc): ampicillin (10 mg), benzylpenicillin (10 U), carbenicillin (100 mg), chloramphenicol (30 mg), oleandomycin (15 mg), ofloxacin (5 mg), rifampicin (5 mg), tetracycline (30 mg), cephalexin (30 mg), erythromycin (15 mg); and resistant to vancomycin (30 mg), kanamycin (30 mg), gentamicin (10 mg), nalidixic acid (30 mg), neomycin (30 mg), oxacillin (10 mg), polymyxin B (300 U), and streptomycin (30 mg). Susceptibility to lincomycin (15 mg) and cephazolin (30 mg) is strain-dependent, the type strain is resistant to lincomycin and cephazolin.

The major menaquinone is MK-6. Major fatty acids are iso-$C_{17:0}$ 3-OH, iso-$C_{15:0}$ followed by iso-$C_{17:1}\omega6$. Polar lipids consisted of phosphatidylethanolamine, one an unidentified aminophospholipid, two unidentified aminolipids, and two or three unidentified lipids. The DNA G+C content of 34.5–34.7% is calculated from the genome sequence.

The type strain of *Profundicola chukchiensis* gen. nov. sp. nov. is strain KMM 9724[T] (= KACC 22806[T]). Isolated from a bottom sediment sampled from the Chukchi Sea in the Arctic Ocean, Russia. The DDBJ/ENA/GenBank accession numbers for the 16S rRNA gene and the whole-genome shotgun sequences of strains KMM 9724[T] and KMM 9713 are OP604014 and LC379507, and JANAIE010000000.1 and JANCMU010000000.1, respectively.

## Supporting information

**S1 Fig. Neighbor-net for novel strains KMM 9724[T] and KMM 9713 and related members of the family *Weeksellaceae* based on MLSA sequences.**
(DOCX)

**S2 Fig. Two-dimensional thin-layer chromatograms of polar lipids of strains.**
(DOCX)

**S1 File. Output files from dbCAN3 meta server.**
(XLSX)

**S2 File. Certificate of deposit and availability of a microorganism in the Collection of Marine Microorganisms (KMM), G.B. Elyakov Pacific Institute of Bioorganic Chemistry, Far-Eastern Branch, Russian Academy of Sciences.**
(PDF)

**S3 File. Certificate of deposit and availability of a microorganism in the Korean Agricultural Culture Collection (KACC).**
(PDF)

## Author Contributions

**Funding acquisition:** Valery Mikhailov.

**Investigation:** Lyudmila Romanenko, Nadezhda Otstavnykh, Valeriya Kurilenko, Peter Velansky, Viacheslav Eremeev, Marina P. Isaeva.

**Methodology:** Lyudmila Romanenko, Nadezhda Otstavnykh, Valeriya Kurilenko, Peter Velansky.

**Project administration:** Valery Mikhailov.

**Resources:** Marina P. Isaeva.

**Software:** Nadezhda Otstavnykh, Viacheslav Eremeev.

**Writing – original draft:** Lyudmila Romanenko, Nadezhda Otstavnykh, Marina P. Isaeva.

**Writing – review & editing:** Lyudmila Romanenko, Valery Mikhailov, Marina P. Isaeva.

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
