## [Decision Letter · Decision Letter 0]

5 Apr 2023

PONE-D-23-01800Description and genome-wide analysis of <profundicola chukchiensis=""> gen. nov., sp. nov., marine bacteria isolated from bottom sediments of the Chukchi Sea</profundicola>PLOS ONE

Dear Dr. Isaeva,

Thank you for submitting your manuscript to PLOS ONE. As your manuscript (ms) currently stands it does not does not fully meet PLOS ONE’s publication criteria and I have decided to label it as in need of "minor revisions". Therefore, we invite you to submit a revised version of the manuscript that addresses the points raised during the review process.

As you will notice from the comments of both referees at the end of this letter, both referees independently agreed that your ms is in need of minor revisions, hence my decision. I would like to push you to fulfil their comments on your rebuttal letter and resubmit the revised version of your ms as soon as possible since I find your work very interesting and worth publishing. One of the referees has some concern regarding the availability of your data at the time of writing so please also attend that particular situations accordingly when resubmitting the ms on the PLOS ONE platform.  

We look forward to receiving your revised manuscript.

Kind regards,

Luis Angel Maldonado Manjarrez, Ph.D.

Academic Editor

PLOS ONE

https://www.frontiersin.org/articles/10.3389/fmicb.2019.02083/full

https://www.biorxiv.org/content/biorxiv/early/2019/01/16/522367.full.pdf

https://assets.researchsquare.com/files/rs-1570742/v1/6e265435-68da-4e74-9de0-42402259da79.pdf?c=1650639680

In your revision ensure you cite all your sources (including your own works), and quote or rephrase any duplicated text outside the methods section. Further consideration is dependent on these concerns being addressed.

“This work was supported by a grant from the Ministry of Science and Higher Education, Russian Federation 15.BRK.21.0004 (Contract No. 075-15-2021-1052).”

5. Please remove your figures from within your manuscript file, leaving only the individual TIFF/EPS image files, uploaded separately. These will be automatically included in the reviewers’ PDF.

Reviewers' comments:

Reviewer's Responses to Questions

**Comments to the Author**

1. Is the manuscript technically sound, and do the data support the conclusions?

Reviewer #1: Yes

Reviewer #2: Partly

2. Has the statistical analysis been performed appropriately and rigorously? 

Reviewer #1: N/A

Reviewer #2: N/A

3. Have the authors made all data underlying the findings in their manuscript fully available?

Reviewer #1: Yes

Reviewer #2: No

4. Is the manuscript presented in an intelligible fashion and written in standard English?

Reviewer #1: Yes

Reviewer #2: Yes

5. Review Comments to the Author

Reviewer #1: A new genus Profundicola chukchiensis gen. nov., sp. nov., was described basend on genome taxonomy.

Only bifurcating tree reconstruction is unlikely to show monophyletic association to the ohter related taxa.

Authors should perform splits decomposition analysis to cleary show the monophyly of the strains described here.

major

1. 16S rRNA phylogeny must be shown using ML after model tests.

2. MLSA using splitstree network must be performed.

minors

1. title was not correspoded between in manuscript and submision system.

2. L8. please put and before the last author name.

3. L27 and etc. please correct the font "K"MM p724T.

4. L32 and etc. one digit for similarity, ANI, AAI, and DDH should be used.

5. L54. m not meter.

6. L107. seawater.

7. L141. 16S rRNA gene sequencing.

8. L282. non-motile.

9. Fig. 4. general title should be added.

10. Table 1. capitalized firts letter of feature. no vertical and horizontal sublines.

Reviewer #2: This manuscript describes the isolation of two Gram-negative bacterial strains, which based on 16S rRNA gene sequence analysis could be assigned as members of the *Weeksellaceae* family. Further analyses provided evidence for the classification of the strains as members of a novel genus, with the two strains described, representing a novel species within this genus. The authors provided a report on the extensive analyses of the strains and their associated genomic features, further providing evidence for the proposed classification. However, there are some aspects the authors need to consider/address:

1. Lines 67-70: The list of the members belonging to the family *Weeksellaceae* needs to be updated – please see LPSN.dsmz.de for a more complete list.

2. Lines 101, 106, 362: For any % solution reported, please indicate what it represents, i.e., v/v or w/v, etc.

3. Line 110: Please indicate whether an artificial sea water (ASW) control was setup for the API analyses, especially since the strains were suspended in ASW as indicated.

4. Line 148: BLAST analysis and EzBioCloud analysis were performed in 2021. Please indicate whether the same results were obtained during a more recent (2023) search? Sequence information in databases is growing at a phenomenal rate – a more recent analysis would be more accurate.

5. Lines 187-189: dbCAN2 has been updated to dbCAN3 – it would be of value to re-run the genome sequences on the updated version in order to confirm whether the outputs are the same or whether the updated version changes the results presented. dbCAN3 output files should also ideally be downloaded and made available via a data repository.

6. Lines 195-196: Please provide the scripts used for the analyses in R – either as supplementary material or made accessible via a data repository.

7. Line 200: Even though the genome sequence accession numbers are provided, neither provide a hit in GenBank. The same applies to ‘OP604014’. Only ‘LC379507’ returns a result in GenBank for a *Flavobacteriaciae* bacterium Ch26. Please update the GenBank entry to correspond to what is being reported in the manuscript and ensure that the data linked to the other accession numbers are released. Not being able to access the genomes, made it difficult to assess whether what was reported in the manuscript is reproducible.

8. Minor: Please correct the spelling of ‘source’ in line 349, and ‘designed’ in line 26 should be ‘designated’.

6. PLOS authors have the option to publish the peer review history of their article (what does this mean?). If published, this will include your full peer review and any attached files.

Reviewer #1: No

Reviewer #2: No

---

## [Author Response · Author response to Decision Letter 0]

18 May 2023

Responses for Reviewer #1: Thank you very much for reviewing our manuscript. We appreciate your valuable comments and suggestions. We have revised our manuscript thoroughly in accordance with your comments.

Comments: A new genus Profundicola chukchiensis gen. nov., sp. nov., was described based on genome taxonomy. Only bifurcating tree reconstruction is unlikely to show monophyletic association to the other related taxa. Authors should perform splits decomposition analysis to clearly show the monophyly of the strains described here.

Major.

1. 16S rRNA phylogeny must be shown using ML after model tests.

Response: Thank you for your important comment. We have made changes as recommended. Please, see revised figure 1 (Line 280).

2. MLSA using splits tree network must be performed.

Response: According to your recommendation we have performed MLSA to clarify the monophyly of the novel strains. The MLSA was conducted using concatenated sequences of the five housekeeping genes; 16S rRNA, atpD, gyrB, recA, and rpoB, which were retrieved from whole genome sequences. We have added the results (Lines 274-276, S1 File).

Minors

1. title was not corresponded between in manuscript and submission system.

2. L8. please put and before the last author’s name.

3. L27 and etc. please correct the font "K"MM p724T.

4. L32 and etc. one digit for similarity, ANI, AAI, and DDH should be used.

5. L54. m not meter.

6. L107. seawater. (L109)

7. L141. 16S rRNA gene sequencing.

8. L282. non-motile.

9. Fig. 4. general title should be added.

10. Table 1. capitalized firts letter of feature. no vertical and horizontal sublines.

Response: Thank you for your comments. We have made changes as per suggested. Please, look for them in the 'Revised Manuscript with Track Changes'. 

Responses for Reviewer #2: Thank you very much for reviewing our manuscript. We appreciate your valuable suggestions. We have revised our manuscript in accordance with your comments.

Comments: This manuscript describes the isolation of two Gram-negative bacterial strains, which based on 16S rRNA gene sequence analysis could be assigned as members of the Weeksellaceae family. Further analyses provided evidence for the classification of the strains as members of a novel genus, with the two strains described representing a novel species within this genus. The authors provided a report on the extensive analyses of the strains and their associated genomic features, further providing evidence for the proposed classification.

However, there are some aspects the authors need to consider/address:

1. Lines 67-70: The list of the members belonging to the family Weeksellaceae needs to be updated – please see LPSN.dsmz.de for a more complete list.

Response: Thank you for your comments. We have updated the list of the members belonging to the family Weeksellaceae as per suggested. Please, see Lines 80-82.

2. Lines 101, 106, 362: For any % solution reported, please indicate what it represents, i.e., v/v or w/v, etc.

Response: Thank you for your comments. We have made changes as per suggested.

3. Line 110: Please indicate whether an artificial sea water (ASW) control was setup for the API analyses, especially since the strains were suspended in ASW as indicated. 

Response: Yes, indeed, all controls were made as required.

4. Line 148: BLAST analysis and EzBioCloud analysis were performed in 2021. Please indicate whether the same results were obtained during a more recent (2023) search? Sequence information in databases is growing at a phenomenal rate – a more recent analysis would be more accurate.

Response: We agree with your comment. We have checked again and corrected the access date. Please, see the Lines 167.

5. Lines 187-189: dbCAN2 has been updated to dbCAN3 – it would be of value to re-run the genome sequences on the updated version in order to confirm whether the outputs are the same or whether the updated version changes the results presented. dbCAN3 output files should also ideally be downloaded and made available via a data repository.

Response: Thank you so much for your important comment. The genome sequences were annotated via dbCAN3, and output files are presented as tables in supplementary (S1 File). We have added this information to the text of the manuscript. Please, see Lines 243-246. We have to say that predicted CAZyme quantities for both genomes obtained by versions 2 and 3 differed very slightly.

6. Lines 195-196: Please provide the scripts used for the analyses in R – either as supplementary material or made accessible via a data repository.

Response: Thank you for your comment. We used R only for the visualization purposes. We analyzed output files from ANI/AAI-Matrix, TYGS platform, eggNOG-mapper v2, and dbCAN2 meta servers, and corresponding tables were imported in RStudio to obtain heatmaps by the pheatmap package according to manual.

7. Line 200: Even though the genome sequence accession numbers are provided, neither provide a hit in GenBank. The same applies to ‘OP604014’. Only ‘LC379507’ returns a result in GenBank for a Flavobacteriaciae bacterium Ch26. Please update the GenBank entry to correspond to what is being reported in the manuscript and ensure that the data linked to the other accession numbers are released. Not being able to access the genomes made it difficult to assess whether what was reported in the manuscript is reproducible.

Response: We apologize for that. Now, we have released all the sequences reported in the manuscript.

8. Minor: Please correct the spelling of ‘source’ in line 349, and ‘designed’ in line 26 should be ‘designated’.

Response: We apologize sincerely for these typo mistakes. They are rewritten in the revised manuscript.

---

## [Editor Report · Decision Letter 1]

5 Jun 2023

Description and genome-wide analysis of *Profundicola chukchiensis* <profundicola chukchiensis="">gen. nov., sp. nov., marine bacteria isolated from bottom sediments of the Chukchi Sea

PONE-D-23-01800R1</profundicola>

Dear Dr. Isaeva,

We’re pleased to inform you that your manuscript has been judged scientifically suitable for publication and will be formally accepted for publication once it meets all outstanding technical requirements.

Kind regards,

Luis Angel Maldonado Manjarrez, Ph.D.

Academic Editor

PLOS ONE

- - - - - - - - - - - - - - - - - - - - - - - - - - - - - - - - - - - - - - - - - - - -

Additional Editor Comments (optional):

Please make the following changes and pay attention to the last comment entitled "*Final comment*".

**Minor changes**

1. Line 67. It says “has been proposed to include genera”, change to “has been proposed to include the genera”

2. Line 74. It says “During a studying the biodiversity of bacteria isolated”, change to “During a study on the bacterial biodiversity”

3. Line 76. It says “were found”, change to “were recovered”

4. Line 77. It says “methods, and results obtained are”, change to “methods; the results obtained are”

5. Line 78. It says “reported in the present study.”, change to “reported in this study.”

6. Line 100. It says “Gram-staining, oxidase, and catalase reactions, and”, change to “Gram-staining, oxidase, catalase reactions, and”

7. Line 162. It says “and sequenced was performed subsequently using”, change to “and subsequently sequenced using”

**Final comment**

Perhaps a small paragraph regarding the phylogenetic relationship found between your proposed novel genera (ie. *Profundicola*) and the poultry pathogen, *Ornithobacterium rhinotracheale* should be added but should be carefully written.

Some colleagues while reading your manuscript (ms) may well argue a “possible” relationship between the two genera, namely a non-pathogenic (*Profundicola*) and a pathogenic one (*Ornithobacterium*). Personally, I find very interesting the phylogenetic relationship found in your ms, but at the same time, the lack of any “ecological-environmental” relationship between the putative origin of isolates belonging to each genera. This raises novel questions regarding the source of infection for poultry: can the sea then also be a reservoir for such pathogenic species? How do birds get the infection? Is there any other relationship -except for the phylogenetic one- between the species of the two genera (as mentioned above either ecologically and/or environmentally)?

You can also avoid including this proposed paragraph but perhaps in the future you and the colleagues involved in the ms might well encounter some criticism based on the points I have tried to raise in this final commentary so it may well be worth to have (or start thinking of) a proper answer at any given time.

I should emphasise that the inclusion or the exclusion of the paragraph I suggest does not affect/change my decision as academic editor on accepting your ms as currently stands.

Best regards,

Dr LA Maldonado

- - - - - - - - - - - - - - - - - - - - - - - - - - - - - - 

Reviewers' comments:

No second round of reviewers were needed since all the first round of comments/suggestions were followed accordingly on their rebutal letter.

---

## [Editor Report · Acceptance letter]

18 Jul 2023

PONE-D-23-01800R1 

Description and genome-wide analysis of *Profundicola chukchiensis* gen. nov., sp. nov., marine bacteria isolated from bottom sediments of the Chukchi Sea 

Dear Dr. Isaeva:

I'm pleased to inform you that your manuscript has been deemed suitable for publication in PLOS ONE. Congratulations! Your manuscript is now with our production department. 

Kind regards, 

on behalf of

Dr. Luis A Angel Maldonado Manjarrez 

Academic Editor

PLOS ONE